# Artificial intelligence-supported diabetic retinopathy screening in Tanzania: rationale and design of a randomised controlled trial

Charles R Cleland [1,2] Covadonga Bascaran,[1] William Makupa,[2] Bernadetha Shilio,[3] Frank A Sandi [4] Heiko Philippin [1,5] Ana Patricia Marques [1] Catherine Egan,[6] Adnan Tufail,[6] Pearse A Keane,[6] Alastair K Denniston,[6,7] David Macleod,[8] Matthew J Burton[1,6]

For numbered affiliations see end of article.

**Correspondence to**
Dr Charles R Cleland;
charles.cleland@lshtm.ac.uk

## ABSTRACT

**Introduction** Globally, diabetic retinopathy (DR) is a major cause of blindness. Sub-Saharan Africa is projected to see the largest proportionate increase in the number of people living with diabetes over the next two decades. Screening for DR is recommended to prevent sight loss; however, in many low and middle-income countries, because of a lack of specialist eye care staff, current screening services for DR are not optimal. The use of artificial intelligence (AI) for DR screening, which automates the grading of retinal photographs and provides a point-of-screening result, offers an innovative potential solution to improve DR screening in Tanzania.

**Methods and analysis** We will test the hypothesis that AI-supported DR screening increases the proportion of persons with true referable DR who attend the central ophthalmology clinic following referral after screening in a single-masked, parallel group, individually randomised controlled trial. Participants (2364) will be randomised (1:1 ratio) to either AI-supported or the standard of care DR screening pathway. Participants allocated to the AI-supported screening pathway will receive their result followed by point-of-screening counselling immediately after retinal image capture. Participants in the standard of care arm will receive their result and counselling by phone once the retinal images have been graded in the usual way (typically after 2–4 weeks). The primary outcome is the proportion of persons with true referable DR attending the central ophthalmology clinic within 8 weeks of screening. Secondary outcomes, by trial arm, include the proportion of persons attending the central ophthalmology clinic out of all those referred, sensitivity and specificity, number of false positive referrals, acceptability and fidelity of AI-supported screening.

**Ethics and dissemination** The London School of Hygiene & Tropical Medicine, Kilimanjaro Christian Medical Centre and Tanzanian National Institute of Medical Research ethics committees have approved the trial. The results will be submitted to peer-reviewed journals for publication.

**Trial registration number** ISRCTN18317152.

### STRENGTHS AND LIMITATIONS OF THIS STUDY

⇒ This study will measure the effect of artificial intelligence (AI)-supported diabetic retinopathy (DR) screening and face-to-face counselling on the proportion of persons with true referable DR who attend follow-up at the central ophthalmology clinic.
⇒ The study benefits from a pragmatic design with broad inclusion criteria and involves the implementation of an AI system into a DR screening programme.
⇒ For practical reasons, it is not possible to mask participants to their arm allocation.
⇒ This study is not powered to investigate whether AI-supported screening increases treatment uptake or improves visual outcomes.

neonatal and nutritional diseases combined as the leading cause of mortality in sub-Saharan Africa (SSA).[1] The rising number of people with diabetes is a major contributor to this and from 2021 to 2045 the number of people with diabetes is projected to rise by 129% in SSA.[2] This is proportionally more than any other region of the world.[2 3]

Diabetic retinopathy (DR) is a common complication of diabetes and is a leading cause of blindness globally.[4] DR is asymptomatic in its early stages and often only begins to affect vision when the disease is at an advanced stage when is much less likely to respond to treatment; therefore, screening is recommended to detect potentially sight-threatening DR and to refer people for assessment and treatment by an ophthalmologist. Early detection and timely treatment of sight-threatening DR can prevent blindness in 95% of people with this stage of the disease.[5]

Several countries have successfully introduced DR screening programmes. For example, in the UK, following the

## INTRODUCTION

By 2030, non-communicable diseases are set to overtake communicable, maternal,

introduction of a national DR screening programme, DR is no longer the leading cause of blindness in working-aged adults.[6] However, in many low and middle-income countries (LMICs) there are no or very limited DR screening services and those that do exist suffer from several challenges.

At 12.3%, Tanzania is estimated to have the highest age-adjusted prevalence of diabetes in adults in Africa[2]; this has increased from an estimated prevalence of 2.3% in 2011.[2] In 2018, the Tanzanian Ministry of Health published guidelines which recommend annual screening of persons living with diabetes for DR.[7]

From 2010 to 2020 the Kilimanjaro Diabetic Programme (KDP) operated as a mobile community-based DR screening service across the Kilimanjaro region of Tanzania.[8] The screening team regularly visited each diabetic clinic and captured retinal photographs from people living with diabetes. The photographs were then graded at a later date (usually after a few weeks) by an ophthalmology resident (specialist doctor in training) at the Kilimanjaro Christian Medical Centre (KCMC) Eye Department (the referral hospital). Patients screened were then contacted with their results via text message or phone call and were given advice on whether they needed to attend KCMC.

Major challenges were faced by the KDP as well as other similar programmes in the African region. First, there are not enough skilled eye care staff. Often the same eye care staff screen, follow-up and then treat patients with sight-threatening disease, placing additional pressure on already overstretched systems. The number of eye care staff currently in the African region is already unable to cope with the clinical demands and is very likely to be overwhelmed by the projected surge in demand for diabetic eye care services over the next 30 years. Second, there is limited quality control and training of staff in grading retinal images, and it is therefore unclear how accurate the DR gradings are within the current screening programmes. And third, in the KDP, it was noted that there were poor rates of follow-up (42%) for those referred with potentially sight-threatening DR from screening to secondary eye care.[8] This meant the KDP was identifying people with potentially sight-threatening DR at risk of sight loss, but less than half of those people attended the central ophthalmology clinic for assessment; this strongly affected the ultimate impact of the programme.

In Tanzania, there is no widely available postal service and particularly older, poorer people have no or limited access to telephones and the internet, making it practically impossible to provide some patients with their results and follow-up advice at a later date. Moreover, through 300 interviews with patients with diabetes registered with the KDP, we found that an understanding that DR can be treated and knowledge of the location of the referral hospital are positively related to adherence to follow-up. The provision of an immediate referral decision with real-time feedback and point-of-screening counselling is likely to improve the rates of follow-up for those referred to the eye clinic with sight-threatening DR.[9 10]

In view of the substantial projected increases in the number of people with diabetes worldwide, particularly in LMICs, there is an urgent need to test, develop and implement effective screening services for DR in order to prevent sight loss from the disease.

An alternative system to screen patients for DR that reduces demands on specialist staff, improves quality control and adherence to follow-up is needed. Advances in artificial intelligence (AI) for healthcare offer a potential alternative.

Ophthalmology is a leading medical specialty in healthcare AI, evidenced by the fact that the first autonomous medical AI device to get Food and Drug Administration (FDA) approval (IDx-DR) was for DR grading.[11] There are several AI systems commercially available for DR grading that have shown promisingly high sensitivities and specificities.[12] Such systems work by automating the interpretation of fundus photographs for DR, thereby reducing the need for trained clinical staff and, with high sensitivities and specificities, could enable the wide availability of high-quality diagnostics within a short time-frame. Additionally, the immediate feedback provided by AI-supported screening software would enable a point-of-screening referral decision and counselling.

The current models of screening for DR do not work optimally in countries such as Tanzania. AI offers a promising solution to the specific problems faced in Tanzania through decreased workforce pressures, improved quality control and potentially improved rates of follow-up for those people with sight-threatening DR. However, in order for this technology to be appropriately and successfully implemented it needs to be tested and evaluated in prospective clinical trials.

AI-supported DR screening offers a potentially higher quality, lower unit cost and more accurate way to screen for DR, which is becoming an increasingly significant public health problem. To our knowledge, this study is the first independent trial in SSA to measure whether the implementation of an AI system into a DR screening programme with a point-of-screening result and face-to-face counselling can increase the proportion of persons with true referable DR who attend the central ophthalmology clinic. The results will contribute valuable evidence for considering the wider implementation of AI technologies for DR screening.

## Objectives

The primary objective of this study is to determine if AI-supported DR screening can increase the proportion of people with true referable DR who attend the central ophthalmology clinic following referral after screening for assessment and treatment. The secondary objectives are: (1) to compare prospective programmatic sensitivity and specificity of AI-supported and standard DR screening for the detection of referable DR and (2) to determine the impact of AI-supported screening on other key screening

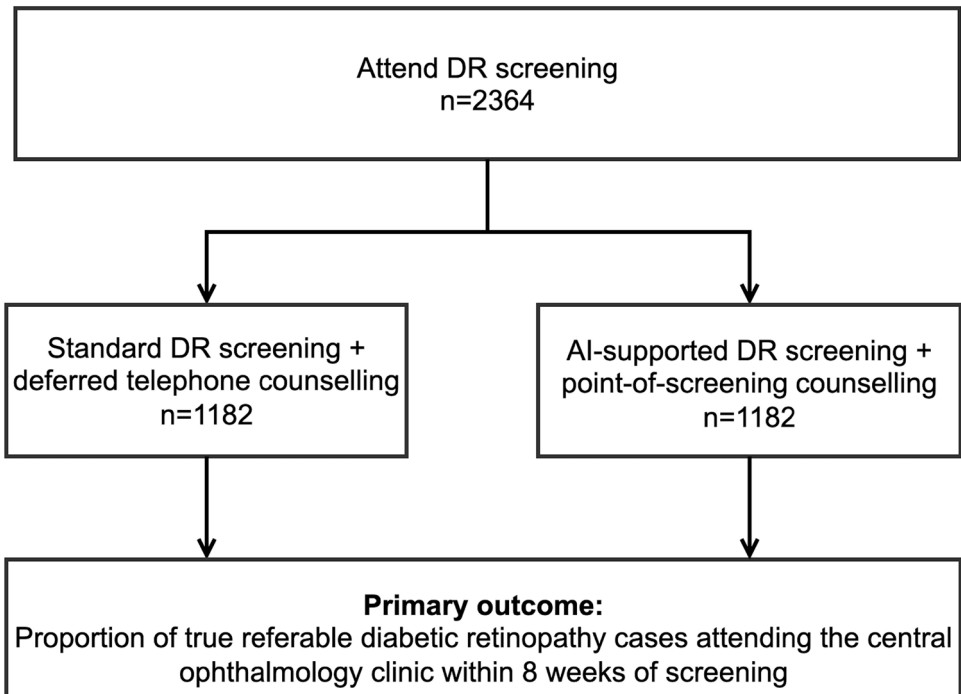

**Figure 1** Overview of the trial. AI, artificial intelligence; QALY, quality-adjusted life year.

programme indicators including false positive referrals, treatment uptake, gradable versus ungradable image capture and to undertake an economic evaluation.

## METHODS AND ANALYSIS
### Trial summary
In an individually randomised controlled trial (RCT), we will test the hypothesis that AI-supported DR screening increases the proportion of people with true referable DR who attend the specialist eye clinic for assessment and treatment following referral after screening.

Persons with diabetes attending screening for DR will be randomised to either the current practice DR screening pathway or the AI-supported DR screening pathway (figure 1). The AI-supported screening arm will include automated retinal image analysis with an immediate referral decision and point-of-screening

counselling. We will assess the proportion of people with true referable DR who attend a follow-up appointment at the central ophthalmology clinic within 8 weeks of screening.

### Trial setting
The trial will be conducted in the Kilimanjaro and Arusha regions of northern Tanzania. KCMC Eye Department is a tertiary referral centre with a full range of ophthalmic specialists including medical retina, vitreoretinal, glaucoma, paediatric, cornea and cataract specialists. As well as fundus cameras, the department has optical coherence tomography (OCT) imaging and both retinal laser and anti-vascular endothelieal growth factor (VEGF) treatments for DR are available. KCMC Eye Department is where patients will be referred after screening if they have referable DR.

**Table 1** Inclusion and exclusion criteria for enrolment into the randomised controlled trial

| Inclusion criteria | Exclusions criteria (any of the following) |
|---|---|
| 1. Adult (18 years and older) living with diabetes and attending diabetes clinics in Kilimanjaro or Arusha region. | 1. Unable or unwilling to consent. |
| 2. Willing and able to give consent. | 2. Less than 18 years old. |
| 3. Agree to be randomised to either AI-supported or standard of care DR screening. | 3. Already attending the central ophthalmology clinic or had a diabetic eye examination in the previous 12 months. |

AI, artificial intelligence; DR, diabetic retinopathy.

## Eligibility criteria

Potential participants will need to meet the eligibility criteria stipulated in table 1. In summary, potentially eligible participants include all persons aged over 18 years with diabetes in the Kilimanjaro or Arusha regions of Tanzania and who attend a diabetic clinic, which is where DR screening occurs. The eligibility criteria are broad to reflect the reality of DR screening in Tanzania.

## Informed consent

Any patients who may be eligible to participate will be given a participant information sheet and its contents will be read out to them by a study team member. They will be asked if they would be willing to participate and it will be explained to the potential participant that depending on which arm of the trial they are randomised to, they will receive their result either on the day of screening or some weeks later.

The patient will then have the opportunity to discuss any questions they might have. If the patient agrees to participate, they will be asked to read and sign or place a thumb print on the study consent form (see online supplemental file 1).

## Clinical assessment

Following assessment of eligibility criteria and the informed consent process, the participant will proceed with the clinical assessment (table 2). In summary, it will include a basic demographic and clinical history, visual acuity testing, blood sugar and blood pressure measurement, followed by fundus photography.

## Intervention and standard of care
### Standard of care

The standard of care is the process by which patients have been screened for DR in the Kilimanjaro DR screening programme for >10 years. After retinal image capture, the photographs will be stored on a laptop and taken back to KCMC Eye Department. The images will be graded by an ophthalmology resident for the level of DR. Persons with referable DR will be contacted by text message or phone call some weeks after screening and advised to attend KCMC Eye Department for assessment with or without treatment. The counselling participants in the standard of care arm receive will follow the same format as the counselling given in the intervention arm of the trial, except that it will be delivered by telephone.

**Table 2** Demographic and clinical data

| Assessment | Details |
|---|---|
| Demographic history | Sex, age, education level, occupation, self-reported tribe/ethnicity, health insurance status. |
| Clinical history | Type of diabetes, duration of diabetes, current diabetes treatment (diet, medications, insulin), smoking history, alcohol consumption, known hypertension, any known diabetes complications, previous heart attack or stroke. |
| Visual acuity | Presenting best corrected VA. VA will be measured using a Tumbling-E logMAR chart with the PEEK VA smartphone application[20] by a trained study team member, for each eye separately. |
| Blood pressure | BP will be measured using a calibrated A&D (A&D Instruments, Abingdon, UK) UA-651 BP monitor.[21] Using the WHO STEPwise approach to NCD risk factor surveillance protocol,[22] three measurements will be taken 1 min apart after the participant has rested for 5 min, with an average of the final two measurements recorded. |
| Random blood sugar | A random blood sugar measurement will be taken using a Contour Next Meter (Ascensia Diabetes Care, UK)[23] and recorded in mmol/L. |
| Fundus photography | One drop of tropicamide 1% will be instilled into each eye separately. After 30 min (to allow adequate pupil dilation) a Topcon NW8 fundus camera will be used to capture two retinal photographs of each eye: one disc centred image and one macular centred image. The retinal images will be captured by a trained medical photographer. |

BP, blood pressure; NCD, non-communicable disease; VA, visual acuity.

## Intervention arm

The intervention in this trial is a complex intervention[13] including AI-supported retinal image analysis for DR grading with immediate feedback and referral advice followed by point-of-screening patient counselling.

There are two components to the intervention:

► AI retinal image analysis and DR grading.
► Face-to-face counselling at the point of screening.

The use of AI to grade retinal images for DR enables point-of-screening, face-to-face counselling as automated analysis provides immediate feedback after image submission, with a referral decision. The screening team member will inform the participant of their DR grade and proceed with the counselling component of the intervention. The point-of-screening counselling will include an explanation of what DR is, the potential to cause vision loss, the referral process and will emphasise that there are effective and safe treatments for DR.

### AI system selection

There are several AI systems commercially available for DR screening.[12] A review process informed the choice of AI system in this trial.[14] In summary, the following criteria were considered: existing regulatory approval (FDA approval or CE marking), ability to work offline, evidence of performance in an African population and the ability to work with different cameras. The SELENA+AI software (EyRIS)[15] was the only system to meet all these criteria and was therefore selected. In a prior published validation study in Zambia the SELENA+AI system had a sensitivity of 92.25% and a specificity of 89.04% for detecting referable DR.[16]

### Counselling component of the intervention

In order to develop a clear, concise and contextually appropriate counselling script to be used to inform participants of their screening outcome, two focus group discussions (FGD) have been held. Participants in the FGDs included persons living with diabetes in the Kilimanjaro region and who are eligible to participate in the trial. A description of the research was provided and the participants were specifically asked what information they would like to be told when receiving their screening result. The counselling component of the intervention will be piloted with persons living with diabetes in the Kilimanjaro region prior to the start of the trial.

To ensure that the same information is shared in a similar form of words in both trial arms, the same counselling script will be used when patients are given their screening result irrespective of their allocation.

### Randomisation
#### Randomisation sequence

A computer-generated randomisation list will be prepared by a statistician at London School of Hygiene & Tropical Medicine (LSHTM). The sequence will be in a 1:1 allocation ratio of standard DR screening to AI-supported DR screening, blocked with a random block size of 4–8.

### Allocation concealment

The randomisation sequence will be concealed in sequentially numbered opaque envelopes. The envelopes will be prepared by an administrator who is independent of all other aspects of the trial. A trained research team member will be responsible for opening the next envelope in numbered sequence and allocating the patient to either the AI-supported or standard of care DR screening pathway.

### Masking

As immediate feedback is provided in the intervention arm versus no feedback at the point of screening in the standard of care arm, it will not be possible to mask participants or screening staff to the allocation. However, the arm the participant is allocated to will only be apparent after randomisation has occurred. The primary outcome is attendance at the central ophthalmology clinic within 8 weeks of screening for those with true referable DR; the research assistant collecting this information at KCMC eye clinic will be masked to the allocation.

### Grading of retinal photographs

All retinal images captured in the trial will be graded by (1) the Tanzanian ophthalmology residents, (2) the AI system and (3) UK-certified graders. This will enable comparison of the performance of the AI system versus Tanzanian ophthalmology residents and against a reference standard (UK-certified graders). All graders will be masked to the other DR gradings.

For the reference standard gradings, the UK-certified graders will undertake feature-based grading and the retinal images will be labelled using a software program (OptoMize, NEC Software Solutions, UK) commonly used in the English National Screening Programme. All images will be labelled by two experienced graders from the English National Screening Programme. The graders will only have access to anonymised retinal images with no patient metadata. Any disagreements will be adjudicated by a third senior grader. All graders involved are subject to the English Diabetic Eye Screening quality assurance process.

Participants noted to have any of the following after grading by the UK-certified graders will be defined as having true referable DR: preproliferative, proliferative DR, referable maculopathy or an ungradable retinal image. Referable maculopathy is defined as the presence of hard exudates or blot haemorrhage within 1 disc diameter of the fovea (centre of the retina) or a circinate or group of exudates within the macula. This definition of referable DR and maculopathy follows the Tanzanian National Guidelines for the Management of DR (published by the Tanzanian Ministry of Health) which is the nationally approved standard of care for Tanzania.

### Follow-up assessment

All participants screened in the trial will be tracked to determine whether or not they attend the central ophthalmology clinic. Any participant recommended to attend a follow-up

**Table 3** Secondary outcome measures and analyses

| Secondary outcome measure | Analysis details |
| --- | --- |
| Proportion of persons who attend the central ophthalmology clinic within 8 weeks of screening out of all those referred | We will measure this outcome, by trial arm, in the same manner as our primary outcome measure. This will be by logistic regression with attendance at 8 weeks the binary outcome and trial arm as the exposure. However, the denominator for this outcome measure will be all persons referred in each trial arm, in contrast to only persons with true referable DR in our primary outcome analysis. |
| Sensitivity and specificity for grading referable DR | All retinal images collected in the study will be graded by the AI system, the Tanzanian ophthalmology residents and the UK-certified graders (reference standard). This will enable us to calculate the sensitivity and specificity of both the AI system and the local Tanzanian ophthalmologists for grading referable DR, relative to the reference standard. |
| False positive cases attending the central ophthalmology clinic | The number of false positive cases attending the central ophthalmology clinic will be compared between trial arms by logistic regression, with false positive cases attending the ophthalmology clinic as the outcome measure and trial arm as the exposure. |
| Number of patients receiving treatment after referral | The study is not powered to detect a difference in treatment uptake but we will undertake an exploratory analysis using logistic regression to compare treatment uptake between the trial arms for those participants who attend the ophthalmology clinic and who are recommended treatment. |
| Number of gradable versus ungradable retinal images | The number of images deemed ungradable will be compared between trials arms by logistic regression with gradable/ungradable retinal image as the outcome and trial arm as the exposure. |
| Time to presentation at hospital | The time to presentation at hospital will be compared between trial arms for all those referred to the central ophthalmology clinic after screening. We will use Cox proportional hazards regression with trial arm as the primary predictor. Survival curves using Kaplan-Meier will be plotted for both trial arms to represent the time to presentation at hospital. |
| Incremental cost per QALY gained | The economic evaluation aims to assess the cost-effectiveness of an AI-supported screening model compared with standard of care model. Cost-effectiveness results will be expressed in terms of incremental cost-effectiveness ratios (ICERs) per QALY gained.<br>A Markov model will be built to undertake the analysis. Both deterministic and probabilistic sensitivity analyses will be performed to test the robustness of the model. Uncertainty in the parameters in the model will be reflected using probability distributions and the overall decision uncertainty will be presented using cost-effectiveness acceptability curves |
| Acceptability, appropriateness and fidelity of AI screening | A process evaluation will be embedded within the trial and will be undertaken using the Medical Research Council guidance.[24] Using semistructured interviews with participants, screening team staff, ophthalmologists and Ministry of Health officials we will assess the acceptability of AI-supported DR screening from different perspectives (patients, staff, policy makers). These qualitative data will be analysed using NVivo according to a condensation of meaning analysis and units of meaning will be coded into central themes.<br>The fidelity of AI-supported screening will be assessed during the trial by random spot checks evaluating the delivery of the intervention. Staff members will be asked to keep a log of any difficulties encountered while delivering the intervention, thereby highlighting implementation barriers and challenges. |

AI, artificial intelligence; DR, diabetic retinopathy; QALY, quality-adjusted life year.

appointment who has not done so within 8 weeks of screening will be considered a non-attender. A follow-up period of less than 8 weeks could bias in favour of the intervention as some participants in the standard of care arm may not receive their screening result until 3–4 weeks after screening. As attendance at the central ophthalmology clinic beyond 8 weeks becomes increasingly less attributable to the DR screening result, it is our view that attendance within 8 weeks of screening is most appropriate.

For participants who attend the ophthalmology clinic we will document what, if any, treatment they are recommended and whether they commence the recommended treatment.

## Lost to follow-up

As the primary outcome is attendance or not at the referral hospital within 8 weeks of screening any persons not attending follow-up will be analysed as such.

## Outcome measures
### Primary outcome measure

The primary outcome will be the proportion of persons with true referable DR who attend the central ophthalmology clinic within 8 weeks of screening out of all those with true referable DR (defined by the reference standard), by trial arm.

All retinal images will be graded by UK-certified graders to provide the reference standard. It will be these gradings that will determine which participants have true referable DR in each trial arm and this figure will be the denominator in our primary outcome analysis.

### Secondary outcome measures

► The proportion of persons who attend the central ophthalmology clinic within 8 weeks of screening out of all those referred, by trial arm.
► Sensitivity and specificity for grading referable DR.

**Table 4** Trial registration summary

| Data category | Information |
|---|---|
| Primary registry and trial identifying number | ISRCTN registry; ISRCTN18317152 |
| Date of registration in primary registry | 2 March 2023 |
| Secondary identifying numbers | |
| Sources of monetary support | British Council for the Prevention of Blindness, Christian Blind Mission and the Sir Halley Stewart Trust |
| Primary sponsor | London School of Hygiene & Tropical Medicine |
| Secondary sponsor | |
| Contact for queries | Dr Charles Cleland; charles.cleland@lshtm.ac.uk |
| Title | Artificial intelligence-supported diabetic retinopathy screening in Tanzania |
| Countries of recruitment | Tanzania |
| Health condition(s) or problem(s) studies | Diabetic retinopathy |
| Intervention(s) | Artificial intelligence-supported retinal image analysis for diabetic retinopathy grading with immediate feedback and referral advice followed by point-of-screening patient counselling |
| Key eligibility criteria | ▶ Adult (18 years and older) living with diabetes and attending a diabetic clinic.<br>▶ Agree to be randomised to either the AI-supported diabetic screening pathway or the standard of care diabetic retinopathy screening pathway. |
| Study type | Randomised controlled trial |
| Date of first enrolment | 8 March 2023 |
| Target sample size | 2364 |
| Recruitment status | Recruiting |
| Primary outcome(s) | Proportion of persons with true referable diabetic retinopathy who attend the central ophthalmology clinic within 8 weeks of screening out of all those with true referable diabetic retinopathy |
| Key secondary outcomes | ▶ The proportion of persons who attend the central ophthalmology clinic within 8 weeks of screening out of all those referred, by trial arm.<br>▶ Sensitivity and specificity for grading referable DR.<br>▶ Number of false positive cases attending the central ophthalmology clinic, by trial arm.<br>▶ Number of patients receiving treatment after referral, by trial arm.<br>▶ Number of gradable versus ungradable retinal images.<br>▶ Time to presentation at hospital.<br>▶ Incremental cost per quality-adjusted life year (QALY) gained. |

AI, artificial intelligence; DR, diabetic retinopathy.

▶ Number of false positive cases attending the central ophthalmology clinic, by trial arm.
▶ Number of patients receiving treatment after referral, by trial arm.
▶ Number of gradable versus ungradable retinal images.
▶ Time to presentation at hospital.
▶ Incremental cost per quality-adjusted life year gained.
▶ Acceptability, appropriateness and fidelity of AI screening.

### Data collection, management, confidentiality and access to data

Demographic and clinical data will be collected electronically in the field using a tablet (Huawei MediaPad T3; Model: KOB-WO9). The data collection forms will be developed using ODK software which is accessible via the tablet. ODK is a free open-source software fully encrypted for collecting, managing and using data in resource-constrained settings and works offline.[17] Data confidentiality will be maintained through restricted access to the data which will be password protected.

The retinal photographs collected in the study will be transferred onto an encrypted laptop after capture. Only authorised members of the study team will have access to these images. Any images transferred out of Tanzania for grading in the UK, in order to provide the reference standard, will be fully anonymised and labelled with internal linking IDs prior to transfer.

### Data and safety monitoring board

The data and safety monitoring board (DSMB) for this trial comprises independent experts in statistics and ophthalmology appointed by the trial steering committee. The DSMB will meet biannually in person or virtually as needed to review the progress of the trial. Any modifications to the study protocol will be reviewed by the relevant ethics committees in the UK and Tanzania and the

DSMB. The DSMB will also monitor any adverse events that occur during the trial.

## Sample size considerations

This study is powered to test the hypothesis that AI-supported DR screening increases the proportion of persons with true referable DR who attend the central ophthalmology clinic following referral after screening.

Our prior evaluation of the DR screening programme showed that the rate of follow-up for persons referred from screening to the ophthalmology clinic was 42%.[8] Using this value, and in discussion with our partners in Tanzania, it was agreed that a 33% increase in compliance with follow-up recommendations would be a significant and programmatically meaningful improvement to the service.

Assuming that 22.5% of persons screened have true referable DR, with 1182 persons screened per trial arm (2364 persons in total), and therefore 266 persons with true referable DR in each arm, we would have a 90% power to detect a 33% uplift (from 42% to 56%) in attendance at the ophthalmology clinic for those persons with true referable DR.

## Analysis plan

The analysis will be by intention to treat and all participant data will be analysed according to their randomisation allocation. However, not all participants randomised will be included in all analyses. For example, our primary outcome analysis includes only those persons with true referable DR as defined by the reference standard gradings.

The data will be analysed and reported according to the Consolidated Standards of Reporting Trials (CONSORT) AI extension guidelines for reporting RCTs.[18] These guidelines are based on the CONSORT 2010 guidelines and have been adapted to produce a set of recommendations for clinical trials evaluating interventions with an AI component.

### Primary outcome analysis: unadjusted analysis

The primary analysis will be a comparison of the proportion of persons with true referable DR out of all those with true referable DR (determined by the reference standard gradings) who attend the central ophthalmology clinic between the standard of care DR screening pathway (control) and the AI-supported DR screening pathway (intervention). The primary analysis of the primary outcome will be by logistic regression model using attendance at 8 weeks as the binary outcome, trial arm as the exposure and all participants who were classed as having referable DR by the reference standard measure will be included in the analysis. The intervention effect of AI-supported screening will be estimated as an OR with a 95% CI.

### Primary outcome analysis: adjusted analysis

We will perform an adjusted analysis, adjusting for measured potential confounders including gender,

education level and distance from the screening clinic to the referral hospital, to account for any potential imbalance between the arms in terms of measured variables (or associated unmeasured confounders).

As attendance versus non-attendance at the specialist eye clinic for participants with true referable DR is the primary outcome measure, missing outcome data due to loss to follow-up will not be an issue. For example, if an individual enrolled in the trial is referred to the central ophthalmology clinic but does not attend the follow-up appointment, they will be considered a non-attender and will be included in the analysis as such; they will not be considered lost to follow-up.

### Analysis of other potential determinants for attendance of true referable DR cases at the eye clinic

Logistic regression models will be used to analyse potential factors that may be associated with the primary outcome (attendance of true referable DR cases at the eye clinic). These potential factors include: age, gender, education level, occupation, distance from screening site to referral hospital, type of diabetic clinic (government funded, private, church based), duration of diabetes, visual acuity, type of diabetic treatment (insulin, tablets) and any previous ophthalmic treatments.

We will initially perform a univariable analysis of all variables, followed by a multivariable analysis adjusting for all potential confounders. Both the univariable and multivariable analyses will be presented.

### Secondary outcome analysis

The secondary outcomes will be analysed as detailed in table 3. Binary outcomes will be analysed using logistic regression and continuous variables with linear regression. The sensitivity and specificity for referable DR of the AI system and the Tanzanian ophthalmology residents will be reported by arm and for all participants.

### Interim analysis

As there is negligible risk to participants an interim analysis is not required and data analysis will take place at the end of the study.

All retinal images collected during the study will be graded by local Tanzanian ophthalmology residents. Any images that are graded as referable DR by the Tanzanian graders but were graded as non-referable by the AI (potential false negatives) will be reviewed by a senior grader. If the senior grader concludes that the participant does meet the referral threshold, participants will be contacted and advised to attend the eye clinic, thereby ensuring minimal risk to patients.

In such circumstances, the initial result from the AI grading will be used in the analysis and would therefore not affect the results.

### Patient and public involvement

Our previous research highlighted the importance of knowledge of the referral process and the fact that DR can be treated as key factors that impact referral adherence,[19]

both of which can be stated during point-of-screening counselling. We have held two FGDs with persons living with diabetes in the Kilimanjaro region, and who would be eligible for DR screening; the participants expressed a preference for a point-of-screening result over a delayed result.

## Ethics and dissemination

Full ethical approval has been received from the LSHTM, UK, and the KCMC and the National Institute for Medical Research ethics committees in Tanzania. Any protocol modifications will be submitted to the relevant parties for review and/or approval. At the end of the study period any patients who still require treatment or follow-up will continue to be treated at KCMC Hospital. The trial has been registered with the ISRCTN clinical trials registry and the LSHTM is the trial sponsor (table 4). Data will be available on request from the corresponding author on completion of the trial. The results of this trial will be submitted for publication in peer-reviewed journals and will be presented at local and international meetings.

**Author affiliations**
[1]International Centre for Eye Health, London School of Hygiene & Tropical Medicine, London, UK
[2]Eye Department, Kilimanjaro Christian Medical Centre, Moshi, Tanzania
[3]Ministry of Health, Community Development, Gender, Elderly and Children, Dodoma, Tanzania
[4]Department of Ophthalmology, University of Dodoma School of Medicine and Nursing, Dodoma, Tanzania
[5]Eye Centre, University of Freiburg Faculty of Medicine, Freiburg, Germany
[6]National Institute for Health and Care Research (NIHR) Biomedical Research Centre (BRC) for Ophthalmology, University College London, Moorfields Hospital London NHS Foundation Trust and Institute of Ophthalmology, London, UK
[7]National Institute for Health and Care Research, Birmingham Biomedical Research Centre, Birmingham, UK
[8]Department of Infectious Disease Epidemiology, London School of Hygiene & Tropical Medicine, London, UK

**Acknowledgements** The authors thank the Kilimanjaro Christian Medical Centre Eye Department for their support with study coordination and implementation. The authors thank the EyRIS team for their support with providing the AI software, in particular Steven Ang. The authors thank the staff at the Eye Screening Centre at the Homerton Hospital, in particular John Anderson and Louis Bolter, for their support with providing the reference standard gradings.

**Contributors** CRC searched the literature, drafted the initial protocol and manuscript and was responsible for conceptualisation and funding acquisition. CRC, CB, WM, BS, FAS, HP, APM, CE, AT, AKD, PAK, DM and MJB contributed to protocol development and revision. CRC, CB, WM, BS, FAS, HP, APM, CE, AT, AKD, PAK, DM and MJB critically revised the manuscript.

**Funding** This research was supported by the British Council for the Prevention of Blindness, Christian Blind Mission and the Sir Halley Stewart Trust. PAK is supported by a Moorfields Eye Charity Career Development Award (R190028A) and a UK Research and Innovation Future Leaders Fellowship (MR/T019050/1). MJB is supported by Wellcome Trust (207472/Z/17/Z).

**Disclaimer** The views expressed are those of the authors and not necessarily those of the NIHR or the Department of Health and Social Care. The funders have had, and will have, no role in the study design, data collection and analysis, decision to publish or preparation of the manuscript.

**Competing interests** CE has received fees from Heidelberg Engineering and Inozyme Pharma. AT has received fees from Annexon, Apellis, Bayer, Genentech, Iveric Bio, Novartis, Oxurion and Roche. PAK has acted as a consultant for Roche, Novartis, Boehringer Ingelheim, Adecco and Bitfount and is an equity owner in Big

Picture Medical; has received speaker fees from Novartis, Gyroscope, Bayer, Thea, Boehringer Ingelheim, Apellis, AbbVie, Alimera, Roche, Genentech, Specsavers, Heidelberg Engineering, Topcon and Santen; has received travel support from Bayer, Topcon and Roche; and has attended advisory boards for Boehringer Ingelheim, RetinAI, Novartis, Apellis, AbbVie and Roche.

**Patient and public involvement** Patients and/or the public were involved in the design, or conduct, or reporting, or dissemination plans of this research. Refer to the Methods section for further details.

**Patient consent for publication** Not applicable.

**Provenance and peer review** Not commissioned; externally peer reviewed.

**ORCID iDs**
Charles R Cleland http://orcid.org/0000-0002-2761-8092
Frank A Sandi http://orcid.org/0000-0003-2073-2041
Heiko Philippin http://orcid.org/0000-0002-5380-6994
Ana Patricia Marques http://orcid.org/0000-0001-8242-7021

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
