## [Reviewer comments · BMJ Open]

ARTICLE DETAILS

TITLE (PROVISIONAL)	Artificial Intelligence-Supported Diabetic Retinopathy Screening in Tanzania: rationale and design of a randomised controlled trial
AUTHORS	Cleland, Charles; Bascaran, Covadonga; Makupa, William; Shilio, Bernadetha; Sandi, Frank; Philippin, Heiko; Marques, Ana Patricia; Egan, Catherine; Tufail, Adnan; Keane, Pearse; Denniston, Alastair; Macleod, David; Burton, Matthew J

VERSION 1 – REVIEW

REVIEWER	Wolf, Risa M. Johns Hopkins University School of Medicine
REVIEW RETURNED	24-May-2023

GENERAL COMMENTS	Provide a description of the AI system chosen, its sensitivity, specificity and diagnosability based on other reports. Followup assessment: provide rationale for a followup period of 8 weeks (v 4 weeks or longer at 12weeks) For inclusion/exclusion criteria, consider adding that a patient should not have had a diabetic eye exam in the last 12 months (If someone has, and their AI screen is positive, and they are referred to ophtho, they may be less likely to go because they just had another eye exam and may already have a scheduled eye care provider clinic visit outside of the one specified here)
--

REVIEWER	Nakayama, Luis Filipe Federal University of São Paulo
REVIEW RETURNED	30-Jun-2023

GENERAL COMMENTS	This study tackles the critical issue of tailoring artificial intelligence applications to suit specific target populations and addresses the need to address bias and unbalanced performance. Here are some suggestions for enhancing the review: 1) Consider collecting self-reported race: To gain a deeper understanding of potential biases in AI applications, it would be valuable to include self-reported race as a data point. This information could help identify any disparities that may exist across different racial or ethnic groups.2) Expand comorbidities information: In addition to systemic hypertension and diabetes, it would be beneficial to collect data on other comorbidities. By considering a broader range of health conditions, the study can provide a more comprehensive analysis of the potential impact of various comorbidities on AI performance.
---

	3) Caution regarding commercial AI systems: Especially when the details of the training/testing dataset and algorithm are not openly available. In addition to evaluating performance through pivotal studies, conducting a thorough bias assessment across subgroups is essential to ensure equitable outcomes. 4) Enhance details about the labeling and adjudication process: To improve the transparency and reliability of the study, it is recommended to provide more information about the labeling and adjudication process. Clarify whether there is only one grader per image or multiple graders with an adjudication process. It is also important to mention whether the graders have access to patient metadata and if any quality control measures are in place within the grading groups. 5) Compare performance with commercial software: To further enhance the study's findings, it would be valuable to develop or fine-tune a screening model and compare its performance and subgroup metrics with the commercial software being evaluated. This comparative analysis can provide insights into the potential benefits or limitations of different AI approaches.
--	---

REVIEWER	Sarohia, Gurkaran University of Alberta, Department of Ophthalmology and Visual Sciences
REVIEW RETURNED	07-Sep-2023

GENERAL COMMENTS	Thank you for your work on this manuscript. The article is indeed very interesting and well written. Being able to use AI technology in underserved countries for screening protocol is going to be very beneficial and the thought of performing a RCT with high quality evidence is fantastic. Please see my detailed thoughts: -Introduction: Line 35 has 2 full stops after diabetes. Introduction has a good argument. -Objective: The primary objective seems fine. The secondary objective, do you need to check the sensitivity and specificity because you'll already be using a software that has perhaps had RCTs on it and is FDA approved. Methods and analysis: Overall sound methodology. Something to consider would be masking. If patients know that it is going to be AI rather than a human interpreting their images, it might make them less likely come to follow-ups due to perception and/or it might only selectively choose patients from a specific demographic who give consent to that. I understand that it is challenging however, it would be something to consider when working on the project and how to tackle this.
--

REVIEWER	Lithoxopoulou, Maria Aristotle University of Thessaloniki, Medicine
REVIEW RETURNED	27-Sep-2023

GENERAL COMMENTS	The abstract is accurate, and balanced and the study design is appropriate to answer the research question The methods described are sufficiently described to accurate the purpose The research ethics (e.g. participant consent, ethics approval) are addressed appropriately. The primary and secondary outcomes are clearly defined The statistics used are appropriate and described fully
--

	The references are up-to-date and appropriate 9. Do the results address the research question or objective? 10. Are they presented clearly?
--	---

REVIEWER	Lai, kunbei Sun Yat-Sen University Zhongshan Ophthalmic Center State Key Laboratory of Ophthalmology
REVIEW RETURNED	06-Oct-2023

GENERAL COMMENTS	It is interesting that the author aims to determine if AI-supported DR screening can increase the proportion of people with true referable DR who attend the central ophthalmology clinic following referral after screening for assessment and treatment. Although the study is feasible and has a certain significance in local areas. However, there are some issues that need to be addressed:  1. This study is not innovative enough, and there have been similar previous articles on AI-based DR Screening to improve referral service uptake in low-resource areas, some even compared different AI systems. The authors should clearly state the innovation of the research to the readers. 2. Page 7: Eligibility Criteria: potentially eligible participants include all persons aged over 18 years with diabetes in the Kilimanjaro region of Tanzania and who attend a DR screening event: Why does the research object include both all persons aged over 18 years with diabetes and persons who attend a DR screening event? The latter part belongs to the former part. Please explain. 3. Other factors such as medical insurance can affect the decisions of the patients, why are they not considered? 4. The images will be graded by at least two residents: In case of missed diagnosis or misdiagnosis. 5. In terms of AI system selection. It is understandable to consider whether the software is applicable to the actual situation of the region (Has it been tested on black people?), but it should be checked whether the accuracy of the software is reliable in advance, or it should be mentioned in the article that the accuracy data of the software has been tested before. So as not to bias the results. 6. Page 7: Exclusion criteria should be more comprehensive and accurate : For example, other local eye diseases, or poorly controlled systemic diseases. 7. Page 9: Referable maculopathy is defined as the presence of hard exudates or blot haemorrhage within 1 disc diameter of the fovea (centre of the retina): In fact, blot haemorrhage within 1 disc diameter of the fovea is meaningless for maculopathy. 8. Why plan to use one disc centred image and one macular centred image rather than the 7-standard field of EDTRS?
---

VERSION 1 – AUTHOR RESPONSE

Reviewer: 1

Risa M. Wolf , Johns Hopkins University School of Medicine

Comments to the Author:

Provide a description of the AI system chosen, its sensitivity, specificity and diagnosability based on other reports.

We have added more details regarding the diagnostic performance of the chosen AI system on pages 8-9.

Followup assessment: provide rationale for a followup period of 8 weeks (v 4 weeks or longer at 12weeks)

We have added further justification for this on page 10.

For inclusion/exclusion criteria, consider adding that a patient should not have had a diabetic eye exam in the last 12 months (If someone has, and their AI screen is positive, and they are referred to ophtho, they may be less likely to go because they just had another eye exam and may already have a scheduled eye care provider clinic visit outside of the one specified here)

Thank you for highlighting this important point. We have added this to the exclusion criteria to (table 1; page 7).

Reviewer: 2

Dr. Luis Filipe Nakayama, Federal University of São Paulo, Massachusetts Institute of Technology
Comments to the Author:

This study tackles the critical issue of tailoring artificial intelligence applications to suit specific target populations and addresses the need to address bias and unbalanced performance. Here are some suggestions for enhancing the review:

1) Consider collecting self-reported race: To gain a deeper understanding of potential biases in AI applications, it would be valuable to include self-reported race as a data point. This information could help identify any disparities that may exist across different racial or ethnic groups.

We thank the reviewer for raising this important point related to algorithmic fairness. The population in the northern Tanzania is homogenous and is almost entirely black African, with a small population of persons of Indian origin. We are collecting self-reported data regarding the participants tribe; this data point has been added this to table 2 (page 7).

2) Expand comorbidities information: In addition to systemic hypertension and diabetes, it would be beneficial to collect data on other comorbidities. By considering a broader range of health conditions, the study can provide a more comprehensive analysis of the potential impact of various comorbidities on AI performance.

We are collecting data on duration of diabetes, smoking history, alcohol consumption, any known diabetes complications and any prior history of heart attack or stroke. These data items are all now listed in Table 2 (page 7). The reviewer raises important considerations about the effect of other health conditions on diabetic retinopathy. However, our grant for this study has a defined question and the study is not powered to measure these variables. Unfortunately due to budgetary constraints and the practicalities of collecting data in the field we are unable to collect more detailed information.

3) Caution regarding commercial AI systems: Especially when the details of the training/testing dataset and algorithm are not openly available. In addition to evaluating performance through pivotal studies, conducting a thorough bias assessment across subgroups is essential to ensure equitable outcomes.

We agree that it is important to be cautious regarding the potential for bias in any AI system, including commercial ones. We therefore conducted a comprehensive review process that led to our choice of the SELENA+ system (this published review is referenced on page 8). A key consideration was that

the selected AI system must have appropriate regulatory approval to be used as a medical device. This was an important consideration for our partner, the Tanzanian Ministry of Health.

This is a pragmatic trial evaluating whether an AI system which is commercially available and ready for deployment and has received appropriate regulatory approvals can lead to programmatic improvements in a diabetic retinopathy screening programme in Tanzania. If a non-commercially available AI system was evaluated, the results would be of limited value to decision makers as the system would be not available to procure and use in clinical practice. Therefore it is our view that evaluating a commercially available AI system with appropriate regulatory approval is important so that the results of the trial have relevance.

4) Enhance details about the labeling and adjudication process: To improve the transparency and reliability of the study, it is recommended to provide more information about the labeling and adjudication process. Clarify whether there is only one grader per image or multiple graders with an adjudication process. It is also important to mention whether the graders have access to patient metadata and if any quality control measures are in place within the grading groups.

We thank the reviewer for their for this comment. The manuscript has been edited to include more detail about the human grading of the images and the adjudication process on pages 9-10 under the section Grading of retinal photographs. In summary the retinal images will be graded independently by two UK certified graders with no access to patient metadata. Any disagreements will be arbitrated by a third senior grader.

5) Compare performance with commercial software: To further enhance the study's findings, it would be valuable to develop or fine-tune a screening model and compare its performance and subgroup metrics with the commercial software being evaluated. This comparative analysis can provide insights into the potential benefits or limitations of different AI approaches.

This is an interesting idea and is something we will consider undertaking with the trial data. However, as this is distinct from the randomised controlled trial, we have not included it in this protocol paper.

Reviewer: 3

Dr. Gurkaran Sarohia, University of Alberta

Comments to the Author:

Thank you for your work on this manuscript. The article is indeed very interesting and well written. Being able to use AI technology in underserved countries for screening protocol is going to be very beneficial and the thought of performing a RCT with high quality evidence is fantastic. Please see my detailed thoughts:

-Introduction: Line 35 has 2 full stops after diabetes.

Thank you, this has been corrected

Introduction has a good argument.

Thank you.

-Objective: The primary objective seems fine. The secondary objective, do you need to check the sensitivity and specificity because you'll already be using a software that has perhaps had RCTs on it and is FDA approved.

As this is a prospective pragmatic trial in a “real-world” clinical programme in Tanzania (where the AI system has not been tested) we think there is value in reporting sensitivity and specificity data. As the previous reviewer has mentioned, most algorithms do not openly publish information about their test and training datasets. When the algorithm is deployed in a new population different from the test/training populations, there is the potential for algorithmic bias. To our knowledge, there is no commercially available algorithm that has included a Tanzanian population in development and there are no publicly available retinal image datasets from Tanzania. It is therefore critically important that we report sensitivity and specificity data in our target population.

Methods and analysis: Overall sound methodology. Something to consider would be masking. If patients know that it is going to be AI rather than a human interpreting their images, it might make them less likely come to follow-ups due to perception and/or it might only selectively choose patients from a specific demographic who give consent to that. I understand that it is challenging however, it would be something to consider when working on the project and how to tackle this.

We thank the reviewer for highlighting some important considerations for our methods that we did consider in the development of this protocol. In the Randomisation section under Masking on page 9, we specifically discuss this. As participants in the intervention arm receive their result immediately after screening versus a delayed telephone result in the standard of care arm it is not possible to mask participants to their allocation. However, participants arm allocation only becomes apparent immediately prior to fundus photography. This occurs after enrolment and after the majority of the data collection has already been completed.

The reviewer raises an interesting question about whether the participants differ in important ways from the average screening population. We may consider this as an exploratory analysis when we publish our results.

We have included some questions about perceptions of AI in the current study, but the reviewer raises some important questions for future research.

Reviewer: 4

Dr. Maria Lithoxopoulou, Aristotle University of Thessaloniki

Comments to the Author:

The abstract is accurate, and balanced and the study design is appropriate to answer the research question

The methods described are sufficiently described to accurate the purpose

The research ethics (e.g. participant consent, ethics approval) are addressed appropriately.

The primary and secondary outcomes are clearly defined

The statistics used are appropriate and described fully

The references are up-to-date and appropriate

9. Do the results address the research question or objective?

10. Are they presented clearly?

We thank the reviewer for their positive comments. There are no specific comments to respond to from reviewer 4.

Reviewer: 5

Dr. kunbei Lai, Sun Yat-Sen University Zhongshan Ophthalmic Center State Key Laboratory of Ophthalmology

Comments to the Author:

It is interesting that the author aims to determine if AI-supported DR screening can increase the proportion of people with true referable DR who attend the central ophthalmology clinic following referral after screening for assessment and treatment. Although the study is feasible and has a certain significance in local areas. However, there are some issues that need to be addressed:

1. This study is not innovative enough, and there have been similar previous articles on AI-based DR Screening to improve referral service uptake in low-resource areas, some even compared different AI systems. The authors should clearly state the innovation of the research to the readers.

We thank the reviewer for their detailed comments. We have edited the final paragraph of the introduction (page 5) to make the innovation of the research more clear to the reader and to highlight the importance of independent evaluation.

Although we note the reviewer's comment, we disagree with the statement that study is not innovative enough. Diabetes and diabetic retinopathy affect people in every demographic and in every country. However, AI systems are developed in a limited number of countries and on a limited number of human populations with datasets typically not available to global developers. There is a growing understanding that algorithms cannot always safely extrapolate to new populations. Therefore, single studies on small populations are not sufficient for confidence in results and studies like this one are necessary.

To our knowledge, this is the first independent trial in sub-Saharan Africa to evaluate whether an AI system implemented into a diabetic retinopathy screening programme can increase compliance with follow-up recommendations. We have referenced (reference 10) the only other study to date conducted in sub-Saharan Africa that had a similar design. This study was conducted by Orbis and evaluated the impact of their own AI system (Cybersight) on follow-up compliance. This trial was not independent as it was run by the organisation who developed and own the AI system. Whilst such studies are important, independent evaluation of systems or evaluation in new populations is critical.

To our knowledge, currently there are no other published studies from low-resource settings evaluating whether an AI-based screening system can improve referral uptake after diabetic retinopathy screening.

This trial is an independent randomised interventional trial and within the broader scope of clinical AI research there are very few such studies. Additionally, this trial also involves implementing an AI system into an active clinical programme. This will help evaluate the critical and to date largely unanswered question of whether an AI system can be successfully implemented into an active diabetic retinopathy screening programme and lead to programmatic improvements.

2. Page 7: Eligibility Criteria: potentially eligible participants include all persons aged over 18 years with diabetes in the Kilimanjaro region of Tanzania and who attend a DR screening event: Why does the research object include both all persons aged over 18 years with diabetes and persons who attend a DR screening event? The latter part belongs to the former part. Please explain.

Thank you for highlighting a lack of clarity in the manuscript

There are many people in Tanzania who have diabetes but do not attend diabetic clinics. It is in the diabetic clinics that people are screened for diabetic retinopathy. Therefore in order to be recruited into the trial it is necessary for participants to have diabetes and to attend a diabetic screening event i.e attend the diabetic clinic.

We have edited some of the text under the Eligibility Criteria (page 7) section to make this more clear.

3. Other factors such as medical insurance can affect the decisions of the patients, why are they not considered?

This is a good point and we are in fact collecting insurance data. This has been added to Table 2 (page 7).

4. The images will be graded by at least two residents: In case of missed diagnosis or misdiagnosis.

The current standard of care in the Kilimanjaro diabetic retinopathy screening programme is that only one ophthalmology resident / trainee grades the retinal images for the level of diabetic retinopathy. This is mainly due to human resource constraints and has been done since 2010 when the programme started. In order that the standard of care arm in this trial is accurately representative, we will therefore follow this approach and have one resident grade all images in the standard of care arm.

5. In terms of AI system selection. It is understandable to consider whether the software is applicable to the actual situation of the region (Has it been tested on black people?), but it should be checked whether the accuracy of the software is reliable in advance, or it should be mentioned in the article that the accuracy data of the software has been tested before. So as not to bias the results.

We thank the reviewer for this request for clarification. The AI system we are using has a prior published validation study from Zambia (a neighbouring African country). We have added a sentence making this more clear and stated the key results of the Zambian validation study on pages 8-9.

6. Page 7: Exclusion criteria should be more comprehensive and accurate : For example, other local eye diseases, or poorly controlled systemic diseases.

We thank the reviewer for this question. There is a common misapprehension that an evaluation of a commercial product in a real world clinical setting should be the same as the selective conditions used for algorithm development or licensing. This approach would substantially limit the benefits of this technology in low resource settings.

This is pragmatic trial in which we are evaluating the impact of implementing an AI system into a “real-world” diabetic retinopathy screening programme. The Kilimanjaro diabetic retinopathy screening programme does not exclude persons with other local eye diseases or poorly controlled systemic diseases from diabetic retinopathy screening, and indeed it would be unethical to do so. We therefore will also not exclude such persons from enrolling in the trial; this will enable us to evaluate more accurately the “real world” impact of using this AI system in an active clinical programme.

7. Page 9: Referable maculopathy is defined as the presence of hard exudates or blot haemorrhage within 1 disc diameter of the fovea (centre of the retina): In fact, blot haemorrhage within 1 disc diameter of the fovea is meaningless for maculopathy.

Thank you for this comment. For this pragmatic trial we are using the definition of referable maculopathy stated in the Tanzanian National Guidelines for the Management of Diabetic Retinopathy which is the approved standard of care in Tanzania. We have rephrased the relevant paragraphs on pages 9-10 to make this more clear.

8. Why plan to use one disc centred image and one macular centred image rather than the 7-standard field of EDTRS?

There is good evidence to support the use of two-field digital fundus photography to screen for diabetic retinopathy. This approach has been demonstrated to achieve good sensitivity (>80%) and specificity (>96%) when compared to seven field photography and has the advantage of being more efficient and cost-effective as well as producing fewer ungradable images. Indeed some diabetic retinopathy screening programmes use only a single photograph per eye.

Mydriatic two-field digital photography is the current standard of care in the Kilimanjaro Diabetic Retinopathy screening programme and two-field digital photography is used in the English Diabetic Retinopathy Screening Programme which informed the design of the Kilimanjaro Diabetic Retinopathy screening programme.

VERSION 2 – REVIEW

REVIEWER	Nakayama, Luis Filipe Federal University of São Paulo
REVIEW RETURNED	16-Dec-2023

GENERAL COMMENTS	The authors successfully addressed my comments. The randomized controlled trial design is now more reasonable and enables validation and generalizability studies in the new population group.
--

REVIEWER	Sarohia, Gurkaran University of Alberta, Department of Ophthalmology and Visual Sciences
REVIEW RETURNED	16-Dec-2023

GENERAL COMMENTS	Thanks for the changes.
-------------------------

REVIEWER	Lai, kunbei Sun Yat-Sen University Zhongshan Ophthalmic Center State Key Laboratory of Ophthalmology
REVIEW RETURNED	20-Dec-2023

GENERAL COMMENTS	The author has resolved the issue I raised.
---